# A pilot-scale comparison between single and double-digest RAD markers generated using GBS strategy in sesame (*Sesamum indicum* L.)

**Pradeep Ruperao[1], Prasad Bajaj[1], Rajkumar Subramani[2], Rashmi Yadav[2], Vijaya Bhaskar Reddy Lachagari[3], Sivarama Prasad Lekkala[3], Abhishek Rathore[4], Sunil Archak[2], Ulavappa B. Angadi[5], Rakesh Singh[2], Kuldeep Singh[6], Sean Mayes[1]\*, Parimalan Rangan[2,7]\***

**1** Center of Excellence in Genomics and Systems Biology, International Crops Research Institute for the Semi-Arid Tropics (ICRISAT), Hyderabad, India, **2** ICAR-National Bureau of Plant Genetic Resources, PUSA Campus, New Delhi, India, **3** AgriGenome Labs Pvt. Ltd., Hyderabad, Telangana, India, **4** Excellence in Breeding Platform, CIMMYT, Hyderabad, India, **5** ICAR-Indian Agricultural Statistical Research Institute, New Delhi, India, **6** Genebank, International Crops Research Institute for the Semi-Arid Tropics (ICRISAT), Hyderabad, India, **7** Queensland Alliance for Agriculture and Food Innovation, The University of Queensland, St. Lucia, Australia

\* sean.mayes@icrisat.org (SM); r.parimalan@icar.gov.in (PR)

**Data Availability Statement:** The raw data in short-read format (2x150 bp) generated for 48 samples for each of the sdRAD and ddRAD

## Abstract

To reduce the genome sequence representation, restriction site-associated DNA sequencing (RAD-seq) protocols is being widely used either with single-digest or double-digest methods. In this study, we genotyped the sesame population (48 sample size) in a pilot scale to compare single and double-digest RAD-seq (sd and ddRAD-seq) methods. We analysed the resulting short-read data generated from both protocols and assessed their performance impacting the downstream analysis using various parameters. The distinct k-mer count and gene presence absence variation (PAV) showed a significant difference between the sesame samples studied. Additionally, the variant calling from both datasets (sdRAD-seq and ddRAD-seq) exhibits a significant difference between them. The combined variants from both datasets helped in identifying the most diverse samples and possible sub-groups in the sesame population. The most diverse samples identified from each analysis (k-mer, gene PAV, SNP count, Heterozygosity, NJ and PCA) can possibly be representative samples holding major diversity of the small sesame population used in this study. The best possible strategies with suggested inputs for modifications to utilize the RAD-seq strategy efficiently on a large dataset containing thousands of samples to be subjected to molecular analysis like diversity, population structure and core development studies were discussed.

## Introduction

Sesame (*Sesamum indicum* L., 2n = 2x = 26), is a member of the Pedaliaceae family, and an oilseed crop that is mainly grown in tropical and subtropical regions. Cultivated sesame is

strategy were submitted in the public repository bearing the submission id: PRJEB60972 or INRP000059.

**Funding:** The authors are grateful to the Department of Biotechnology, Government of India for funding (16113200037-1012166). The URL of the funder is www.dbtindia.gov.in. The funders had no role in study design, data collection and analysis, decision to publish, or preparation of the manuscript. The authors are thankful to AgriGenomics Pvt Ltd (Hyderabad, India) for generating the RAD sequencing data using both sdRAD and ddRAD protocols.

**Competing interests:** The authors have declared that no competing interest exist.

known to be domesticated in the Indian subcontinent [1], although cultivated worldwide in tropical regions. The major producers of sesame were Africa and Asia, with India being the largest producer, although not for the highest productivity (FAOSTAT 2019) [2].

The productivity of sesame in India is low compared to other sesame-producing countries and crop productivity can be improved with genetic improvement by utilizing existing genetic resources [2]. The affordability of next-generation sequencing (NGS) and computational tools have boosted the availability of the sesame reference genome [3–5] and pan-genome assembly [6], and had led the development of the genetic markers that were crucial for various research activities in sesame. Compared to gel-based experiments to discover genetic markers, the high-throughput sequencing-based method had accelerated genome-wide genetic marker development and increased the accuracy of allele calling. One such approach is RAD-seq, which is often applied for genome-wide SNP identification in large genomes through the generation of a reduced representation of the genome and direct sequencing of that representation [7]. This method is relatively low-cost and high-throughput [8]. This technique uses one or two restriction enzymes to digest the whole genome into shorter fragments. Then adaptor primers were ligated and used to amplify a subset of the genome (containing the recognition sequences of the restriction enzymes at their 5' and 3' ends), which is subject to DNA sequencing using the NGS platform. RADseq has been widely used in plants [9–11]. RADseq was further modified to use two restriction enzymes and called ddRAD-seq to have a higher density of sequence representation [12].

It is critical to evaluate the genetic diversity of the available sesame population using molecular tools, preferably DNA-based markers to overcome the environmental influence in phenotype-based diversity assessment. This is especially required when the genotypes are in tens of thousands wherein the phenotyping for all the accessions in a homogeneous environment is nearly impossible. Sequence-based markers have the additional advantage that the genic region variability can be used to associate with the functional variability assessment. Evaluating the genetic diversity of sesame accessions will provide information about how best to use sesame germplasm in a breeding program to accelerate crop improvement. Single nucleotide polymorphism (SNPs) as a molecular marker analysis is one of the most useful methods of investigating the genetic diversity of crop plants [13]. An effective core collection that can capture the maximum genetic diversity of germplasm using the SNP markers would efficiently reduce the number of accessions for phenotypic assessment with only the core collection [14]. Therefore, the assessment of genetic information in different dimensions should be considered when constructing a core collection [15].

A genomic sequence of each sesame sample has equivalent importance for assessing diversity patterns. A reduced representation of the sequence (sdRAD or ddRAD), molecular markers such as SNPs, k-mer signatures are the most helpful genetic resources to estimate the genetic diversity of the sesame population. In this study, we assess the genetic diversity in the 48 sampled collections and aim to identify the representative genotypes that capture the maximum diversity. The sesame diverse samples facilitate the efficient exploration of genetic diversity in germplasm resources. The pilot project with 48 sampled populations would be a useful approach for testing the effectiveness of the large sample collection. In this study, we applied both sdRAD-seq and ddRAD-seq to explore the genetic diversity in sesame sample collection and to identify the suitable approach, by comparing these two.

## Results

### K-mer analysis

**Define the core content of the genome..** The ddRAD-seq and sdRAD-seq tags for 48 sesame samples were sequenced with the Illumina sequencing platform. These 48 samples were selected based on the preliminary phenotypic diversity information for various desired traits as mentioned in the S1 Table in S2 File. The RAD sequencing generated 191.8 million paired-end reads, with a mean of 1.9 million reads per sample (**S1 Table in S2 File**). The ddRAD-seq data of 48 samples were used for k-mer analysis by splitting the sequence reads into k-mers with the count of the resulting k-mer sequence for each sample ranging from 1.3 million (sample B46) to 5.5 million (sample Z37) k-mers with an average production of 2.8 million k-mers (Fig 1A). On the k-mer comparison between the samples, the majority 64.8% (27.6 million) of the k-mers are unique to the sesame samples and the remaining 35.2% of k-mers are shared k-mers between the samples, indicating the level of commonness between the samples. This underscores the representative diversity of the genotypes chosen for the study. For example, 71,455 k-mers are common among the 20 samples, and 35,638 common k-mers were reported between 40 samples. This indicates the shared k-mers decrease as the sample number increases. The cumulative curve of k-mers count reaches the plateau at sample 44 and then gains a higher number of k-mers due to the more unique k-mers present in the remaining four wild samples (N74, I58, Z65, and Z28) (Fig 1B).

K-mers common to all 48 samples were considered as the core k-mers (50,884) (of these 48.6% are genic and 51.4% are intergenic) and k-mers absent in any of the sesame samples were considered as variable k-mers (42.6 million) (of these 3.5% are genic). Based on the abundance of the k-mer in the 48 samples, variable k-mers were categorized into groups with five samples each, of which 90% of variable k-mers contribute from 1–5 samples, which decline till the 36–40 samples with 0.3% and inclines to 0.5% and 0.6% for 41–45 and 46–47 groups of samples, respectively (**S2 Table in S2 File**). This indicates the possibility of these k-mers being softcore k-mers (as these k-mers are absent in one of the samples) due to the reduced (RAD) sequence representations.

A single sketch of k-mers was constructed from the collection of k-mers in the reads and compared to the sketch database. The k-mer based genetic distance between each pair of samples shows that Z28 was the most distinct sample followed by the four more samples (J10, N74, Z37 and Z65) having high genetic distance (Fig 1C). Of these five samples, three samples were wild types (N74, Z28, and Z65) which also have the most distinct k-mers compared to the other sesame samples.

### Gene level k-mer sequence validation

The ddRAD-seq data for 48 samples were mapped to the reference and the gene variability was assessed based on the read mapping. A total of 290 genic RADs (cRADs) were commonly present in all 48 samples and 26,668 genic RADs (vRAD) show variability with genic RADs absent in the number of samples ranging from one to 47 samples. Among the vRAD, 4.1% (1,118) and 0.9% (251) respectively, were uniquely present (genic RAD present in only a single sample) and uniquely absent (genic RADs absent in only one sample). Based on the number of overall vRAD and uniquely present vRAD, eight samples were found to be highly variable from the rest 40 samples. Of these four are wild samples (Z28, Z65, I58 and N74) (**S3 Table in S2 File**). The major vRADs were reported from Z65 (1,395) and Z28 (maximum of uniquely present vRAD 636) (Fig 2A), indicating their diverse nature among the samples (representation) studied.

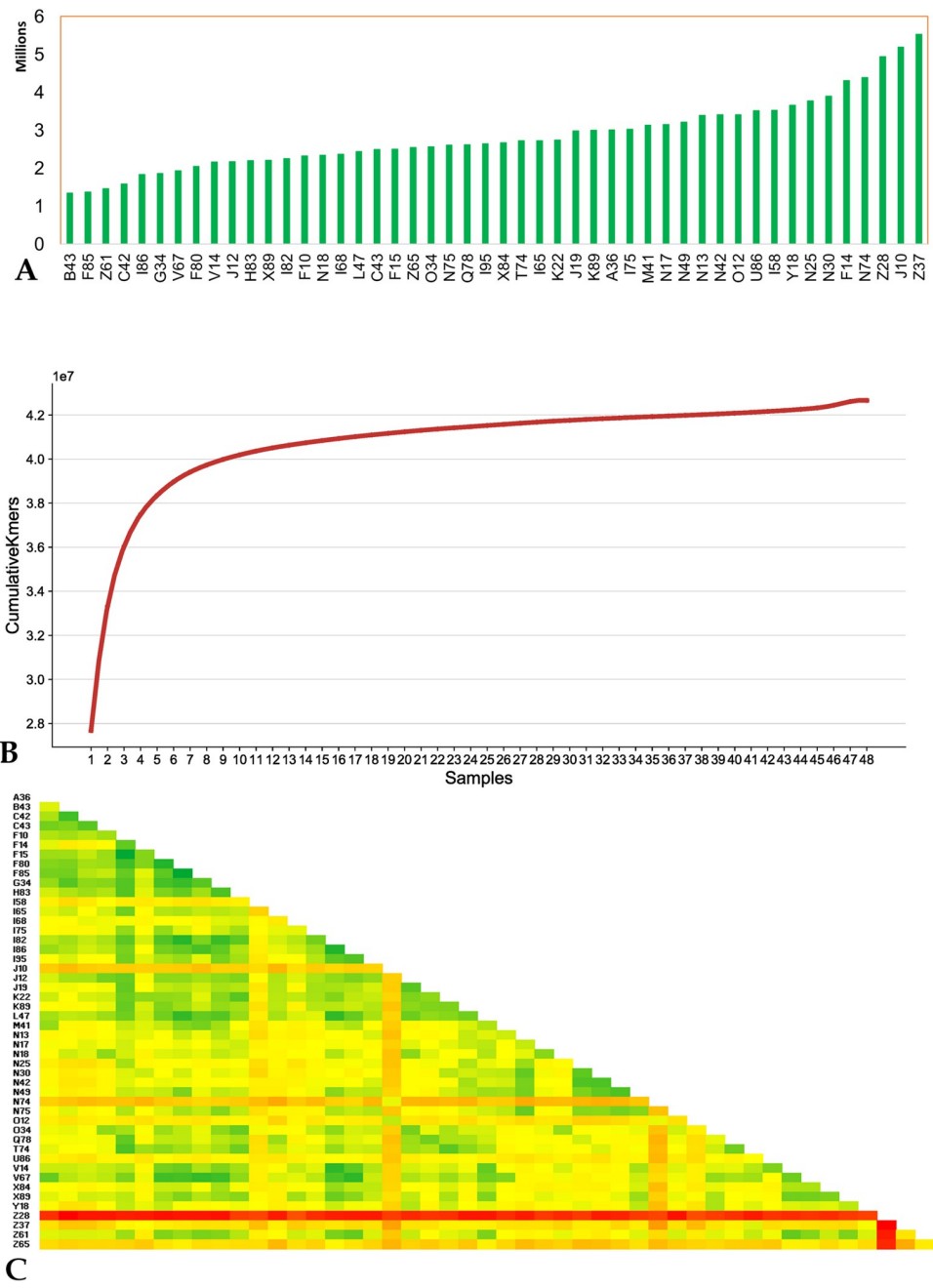

**Fig 1.** K-mer analysis in sesame ddRAD-seq data **A**. distinct k-mer count in each sesame sample **B.** cumulative k-mer count in the sesame 48 sample population **C.** k-mer based mash genetic distance distinct between the sesame samples (colour scale: minimum as green, mean as yellow and maximum as red).

As a possible representative sample subset, these samples capture maximum genetic diversity with a minimal number of genotypically redundant accessions from the sesame population. The k-mers of these samples from the above analysis will assist as a digital signature for the representative samples of the sesame population. In addition to the conserved k-mers (common to all samples), each highly variable sample (X89, I58, V67, Y18, Z28, Z65, N49 and

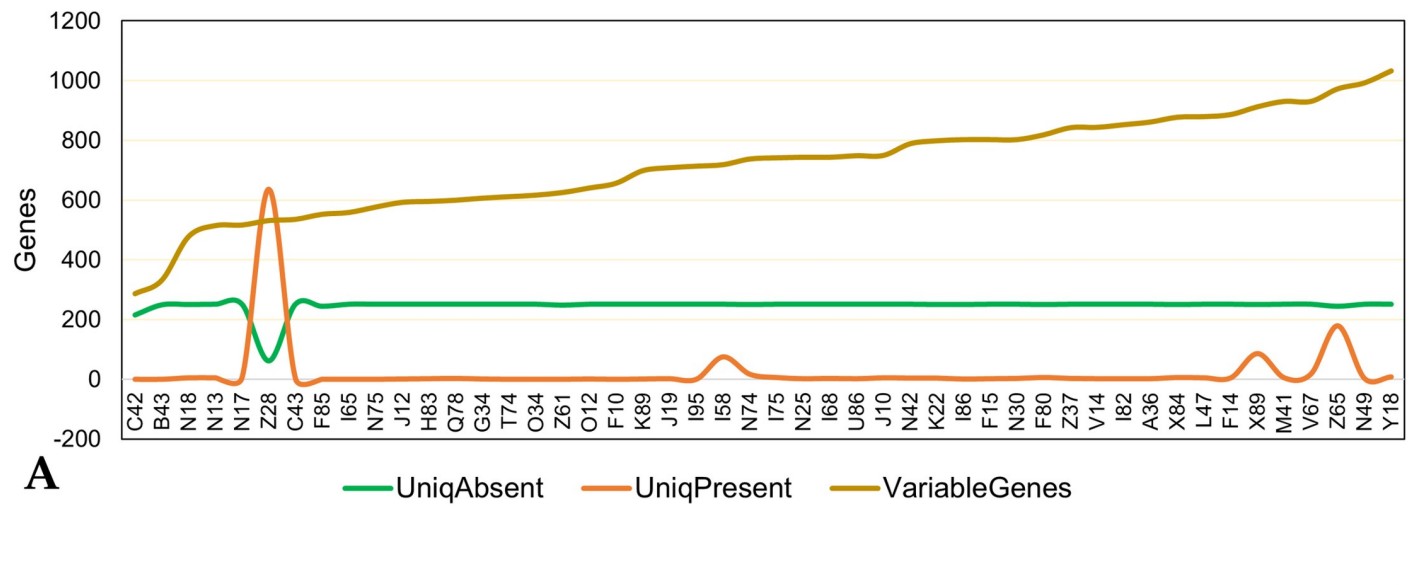

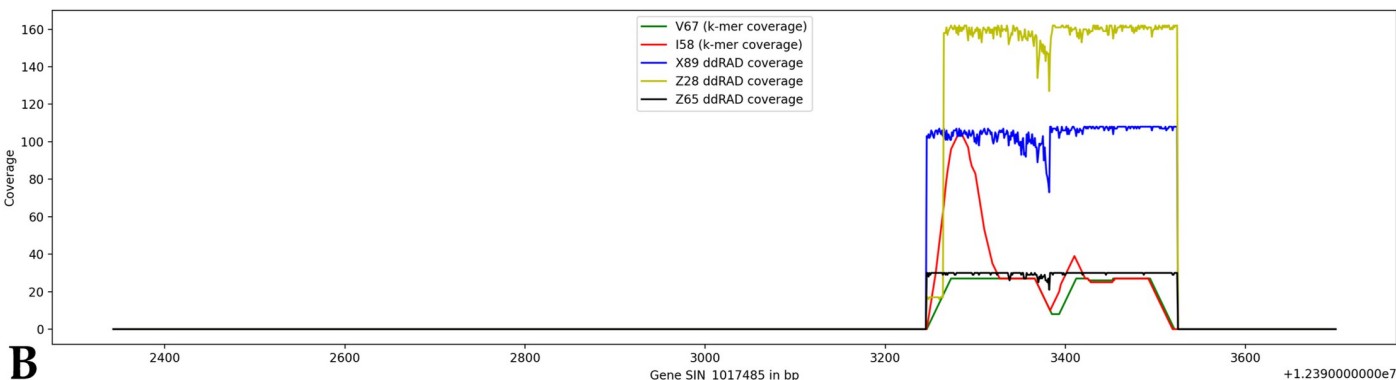

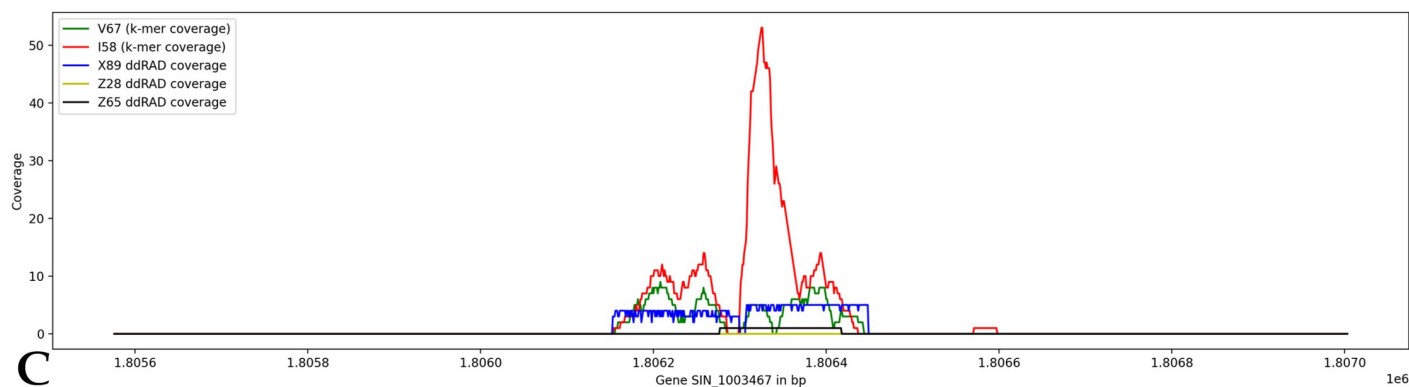

**Fig 2. A.** The ddRAD-seq data-based gene variations **B.** a common gene present in all the sesame samples showing the ddRAD-seq read coverage in five samples **C.** a variable gene showing the ddRAD-seq read from four samples and missing sequence representation in Z28 sample.

N74) from the sesame population set has an average of 3.2 million variable k-mers that hold the maximum diversity compared to the remaining 40 samples (2.7 million k-mers on average from 40 samples) (**S4 Table in S2 File**).

The 99.6% and 89.8% of k-mer defined representativeness and variable sequence supporting the ddRAD-seq mappings to genes, indicates the level of consistency of the sequence reads mapped to the reference genome.

## RAD data analysis and variant calling

The RAD sequence data (sdRAD-seq and ddRADseq) were quality filtered (Q20) and the quality passed reads of each accession were mapped to the sesame reference genome assembly [5]. The filtered reads were aligned with more than 99% of the mapping rate for both sdRAD-seq and ddRAD-seq sequence data (**S5 Table in S2 File**).

A total of 57.3 million ddRAD tag reads (with mean 1.1 million reads per accession) and 6.1 million sdRAD-seq (with a mean of 128,779 per accession) sequence reads were mapped to the reference genome assembly (**S5 Table in S2 File**). On average, the ddRAD-seq data spans 1.3 Mb of the reference genome, whereas sdRAD-seq data spans 1.5 Mb of the reference genome (**S6 Table in S2 File**) (Fig 3). The higher the genome representation, the more possible variants are expected. The sdRAD-seq data had less sequence read representation compared to the ddRAD-seq data. From the sdRAD-seq and ddRAD-seq mapped reads, the SNPs were called and filtered with minor allele frequency > = 0.01 and SNPs were present in 70% of 48 samples, which retained the 13,136 and 27,604 SNPs from sdRAD-seq and ddRAD-seq datasets (**S7 Table in S2 File**), respectively.

We compared the sesame sample allele frequencies between sdRAD-seq and ddRAD-seq in two ways, first individual RAD datasets were analysed separately, and later the combined data was analysed.

The overall distribution of allele frequencies between both data sets, ddRAD-seq and sdRAD-seq, was similar (Fig 4A and 4B). When the ddRAD-seq and sdRAD-seq SNPs were analysed separately, the mean major allele frequency was marginally higher in sdRAD (0.94) than in ddRAD (0.93). For the sample, the mean depth of 18.5 and 149.8 respectively, was reported for sdRAD-seq (maximum of 52.3 and minimum of 3.3) and ddRAD-seq (maximum of 438.3 and minimum of 21.7) (Fig 4C and 4D). The mean depth per loci varies as 19.1 and 144.6 for sdRAD-seq (minimum of 1.9) and ddRAD-seq (minimum of 2.6), respectively. In addition to the higher mean depth per loci and per individual, more SNPs with alternative alleles were reported from ddRAD-seq (average 3,255 SNP/sample) compared to the sdRAD-seq (average 1,019 SNPs) data set. Among the 48 sesame sdRAD-seq samples, I58, Z28, and Z65 have more (than 2000) SNPs with alternative (non-reference) alleles. Whereas in ddRAD-seq dataset, Z28 and Z65 have (more than 5000) SNPs with non-reference alleles (Fig 4E and 4F). Thus the ddRAD-seq dataset provided more genetic information than the sdRAD-seq dataset.

The sdRAD-seq and ddRAD-seq datasets shared 34 common SNPs which indicates the restriction site *ApeKI* is close enough with either *SphI* or *MlucI* restriction site on the reference assembly. These common SNPs were distributed on Chr1 (4), Chr2 (1), Chr6 (1), Chr7 (5), Chr10 (9), Chr13 (3) and 11 SNPs on Scaffold00491 alone (**S8 Table in S2 File**). The common SNP from both datasets may also be due to the presence of adjacent multiple restriction sites in the sesame reference genome assembly. For example, on chr1 at position 1,177,675 has the *Sphl* followed by the *ApeKl* restriction sites allowing to map sdRAD-seq and ddRAD-seq datasets and able to call the common SNPs between both the datasets (**S1 Fig in S1 File**).

## Characterization and annotation of SNPs

The SNP call from both sdRAD-seq and ddRAD-seq datasets were combined for a total of 40,706 SNPs reporting from the 48 sesame samples (Fig 5). The SNPs were annotated to evaluate the impact and measure the effect of identified SNP on the genes. The distribution of SNPs

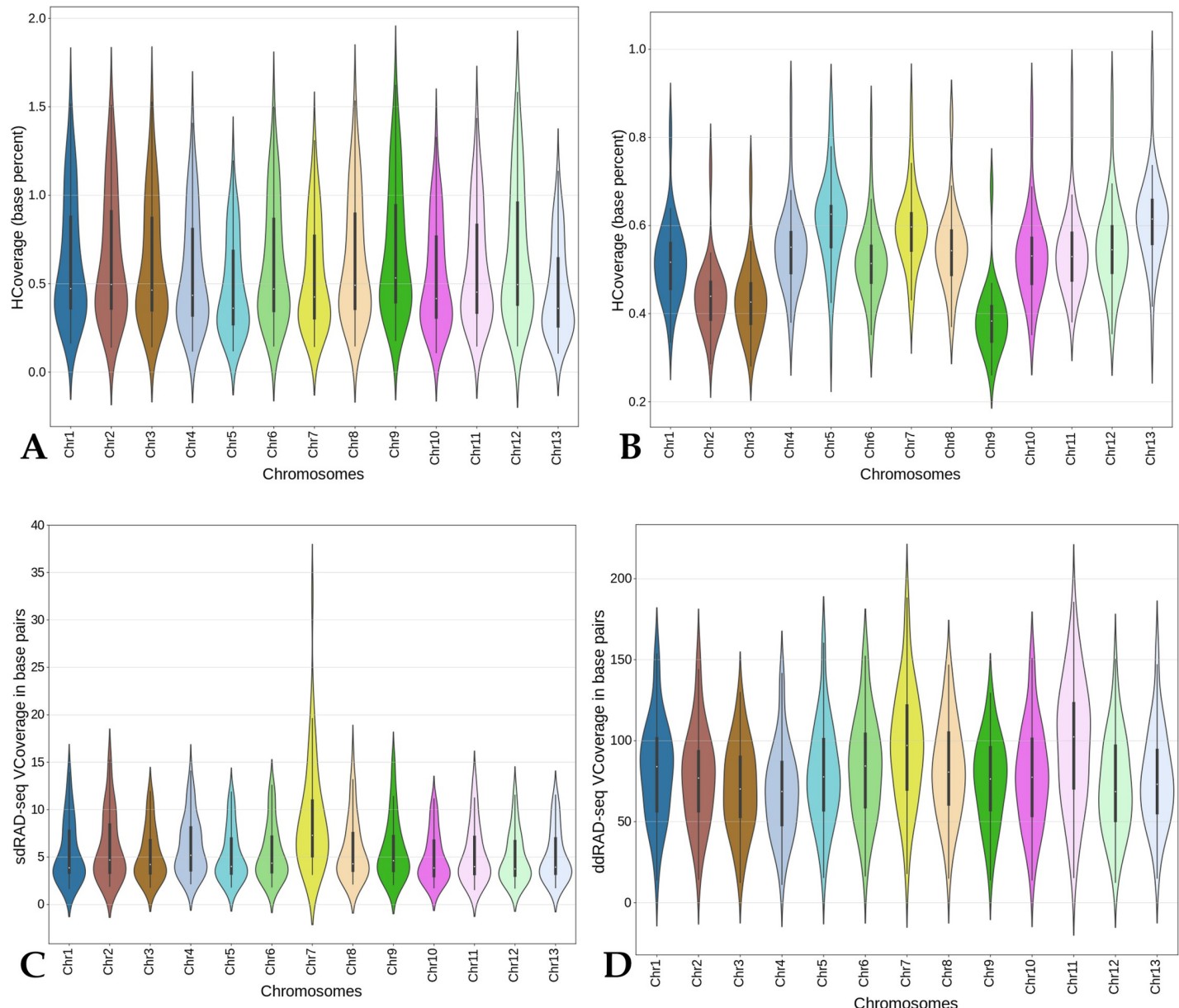

**Fig 3.** The RAD-seq reads mapping spanning the reference genome assembly coverage for **A.** sdRAD-seq **B.** ddRAD-seq and vertical coverage (read depth) for **C.** sdRAD-seq and **D.** ddRAD-seq data.

across the genome shows a low density in the telomere regions of each chromosome compared to the centromere regions (Fig 3). The SNPs abundance in decreasing order were intergenic, exon and intronic genic regions with their proportions of 44.9%, 25% and 14.3%, respectively. A moderate number of SNPs were reported from upstream (7%) and downstream (5.1%) regions compared to the SNPs from 3'UTR (1.7%) and 5'UTR (1.7%). More SNPs were detected in the exon than introns.

Based on the nucleotide substitutions, the combined SNPs identified in the sesame genome were classified into transitions (Ts) (A/G and C/T) and transversions (Tv) (A/C, A/T, G/C, G/T). A total of 24,876 transitions and 14,208 transversions were detected, with a transitions/

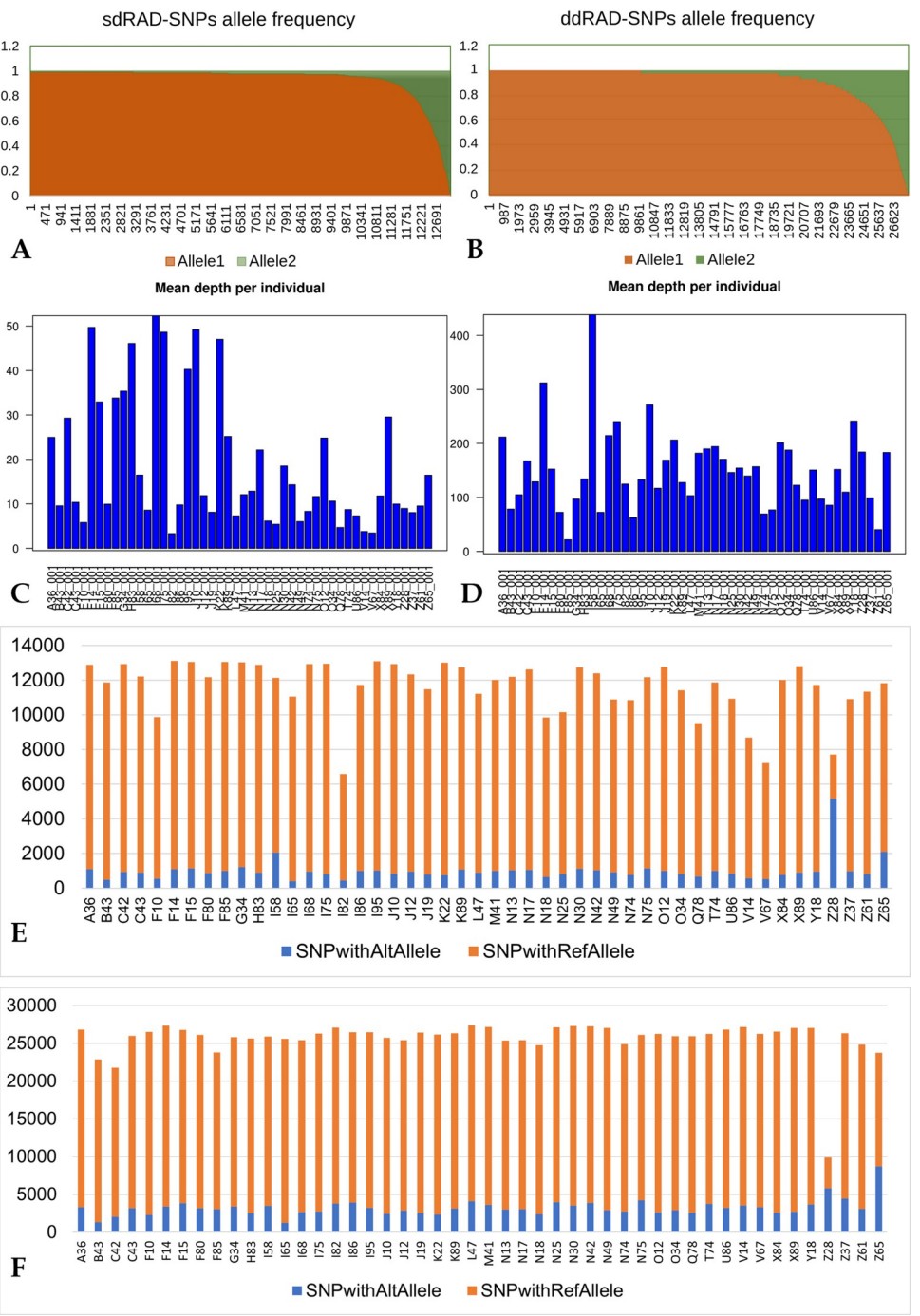

**Fig 4.** The sdRAD-seq and ddRAD-seq data comparison **A** and **B** allele frequency, **C** and **D** mean depth of sequence reads per sample, **E** and **F** plot the SNP count per sample with reference allele and alternative allele for sdRAD and ddRAD, respectively.

transversions (Ts/Tv) ratio of 1.75. The transition frequency of C/T was found to be higher than G/A, as the usual pattern reported earlier in *Coffea arabica* L [16]. The transversion frequency of G/T was greatest followed by C/A and the least frequency detected is the T/A type of transversion (Fig 5C). The maximum of Ts and Tv were identified in intergenic regions with 11,727

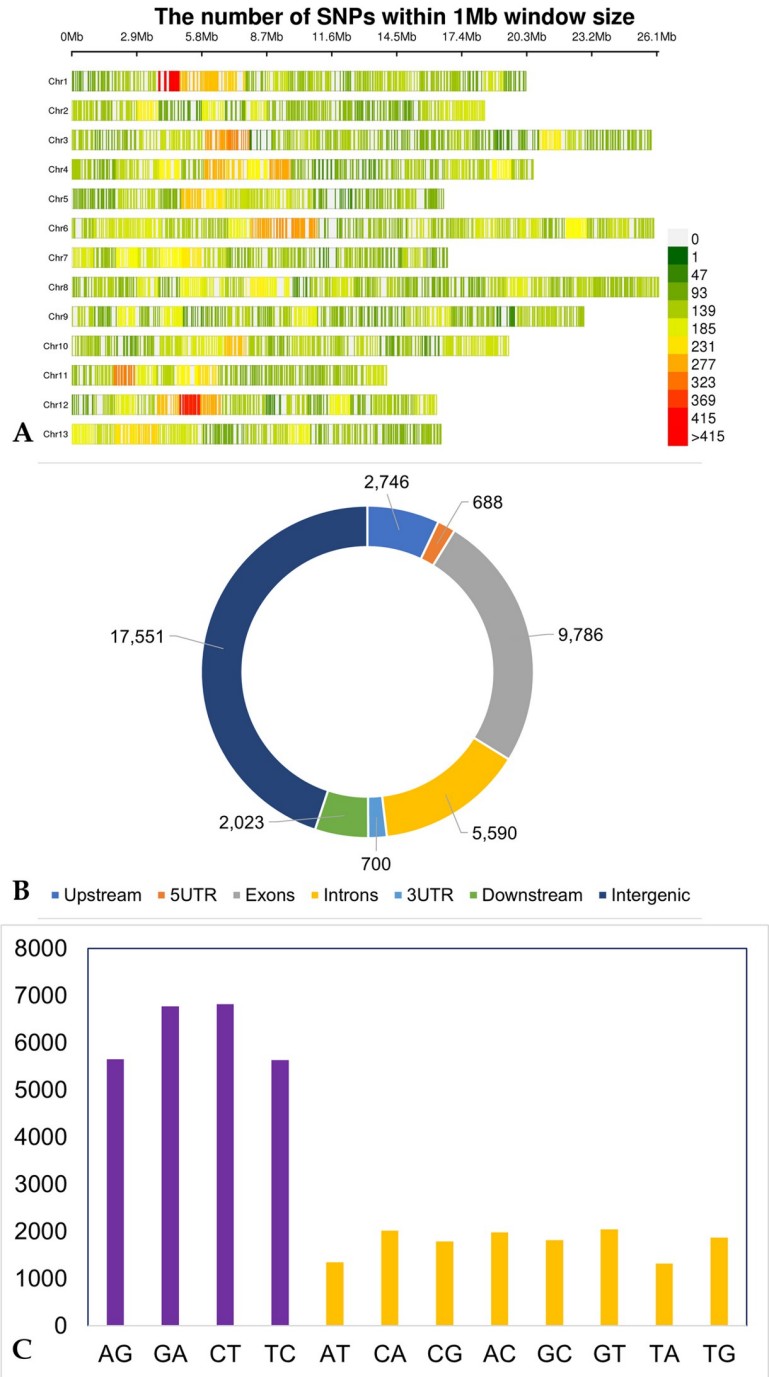

**Fig 5. A.** The genome wide combined (sdRAD-seq and ddRAD-seq) SNP calls density **B.** annotations and **C.** Ts (purple) and Tv (yellow) count in the sesame samples.

and 5,824 respectively. Within the genic regions, the exons have the most Ts (6,074) and Tv (3,712) and UTR regions have the least Ts (789) and Tv (599) nucleotide substitutions. The other regions include the downstream (1,240; 783), intron (3,358; 2,232) and upstream (1,688; 1,058) have identified the moderate level of Ts and Tv's. An average of 23,784 and 13,469 Ts

and Tv were reported in the 48 samples respectively. F14 and Z28 samples recorded the most (25,697 and 14,691) and least (10,801 and 6,763) Ts and Tv SNPs (**S9 Table in S2 File**).

The heterozygosity analysis showed that the observed heterozygosity (Ho) for N75, L47, Z37, Z28 and Z65 samples exhibit a higher value than the expected heterozygosity (He), indicating that these samples are interbreeding. While for other samples the Ho is lower than He (in the range of 0.01 to 0.09), indicating that inbreeding (isolation) is occurring among those populations. Additionally, the inbreeding coefficient (*F*-value) for N75, L47, Z37, Z28 and Z65 samples has negative values indicating the low level of inbreeding compared to the higher inbreed value for the remaining samples indicates the higher gene flow between those populations, supporting the heterozygosity analysis (**S10 Table in S2 File**).

## Sesame genetic diversity

The diverse sesame accessions were inferred for the genetic relationships by constructing a neighbor-joining phylogenetic tree using the combined RAD-SNPs. The analysis revealed three major clusters, and each cluster was further divided into sub-groups (Fig 6). Cluster I has the sesame accessions mostly originating from India (mainly southern states of India), with the exception of I82 samples originating from Nepal. Cluster II has samples originating from Singapore, Japan, USA and the Philippines, and finally, cluster III samples originated from multi nations, such as Singapore, India, and Bangladesh. The four wild samples were distributed between cluster I (N74) and cluster III (I58, Z65 and Z28). The wild samples from cluster III, were genetically more distinct than the N74 wild sample from cluster I. The order of genetic distance (branch length) within the wild accessions was Z28(184) > Z65(100) > I58 (59) > N74(48), indicating these samples were more distinct from the elite samples. The PCA analysis further supports the above genetic relationship between the wild and elite sesame samples collected at different geographical origins. The elite sesame samples were grouped as a single cluster, with a wild sample (N74) close enough and the other three wild samples dispersed away from the elite samples group.

## Overlapping the diversity variables

The eight samples with high diverse k-mers also carried gene level variations (gene presence and absence). In comparison with the other parameters, such as heterozygosity, genetic distance and k-mer based mash distance; our study shows that the four samples are in overlap with the diverse samples predicted from the kmer analysis. Among the four, two samples (Z65, and Z28) are highly heterozygous and relatively distinct to the other samples (Table 1).

## Discussion

The different marker systems are available to reveal the population structure and diversity for the crop improvement program. The sesame markers developed in the earlier studies include the random amplified polymorphic DNA [17], amplified fragment length polymorphism (AFLP) [18], simple sequence repeat (SSR) [19, 20], single nucleotide polymorphisms (SNPs) [20, 21], specific locus amplified fragment sequencing (SLAF-seq) [22]. In this study we used the SNP calling to investigate the diversity in sesame germplasm, a pilot project to assess the diversity in the 48 sub-sampled accessions. The number of SNP markers reported in earlier studies varies with the marker system used and the number of accessions used. For example, Wei et al and Cui et al reported more markers generated with SLAF-seq and whole genome sequence data in large population sizes [21, 22]. RAD sequencing is a reduced representation used for a wide range of applications such as for the construction of genetic maps [23], assessing diversity [24], developing indel [25] and SNP markers [24]. In this study, the sdRAD and

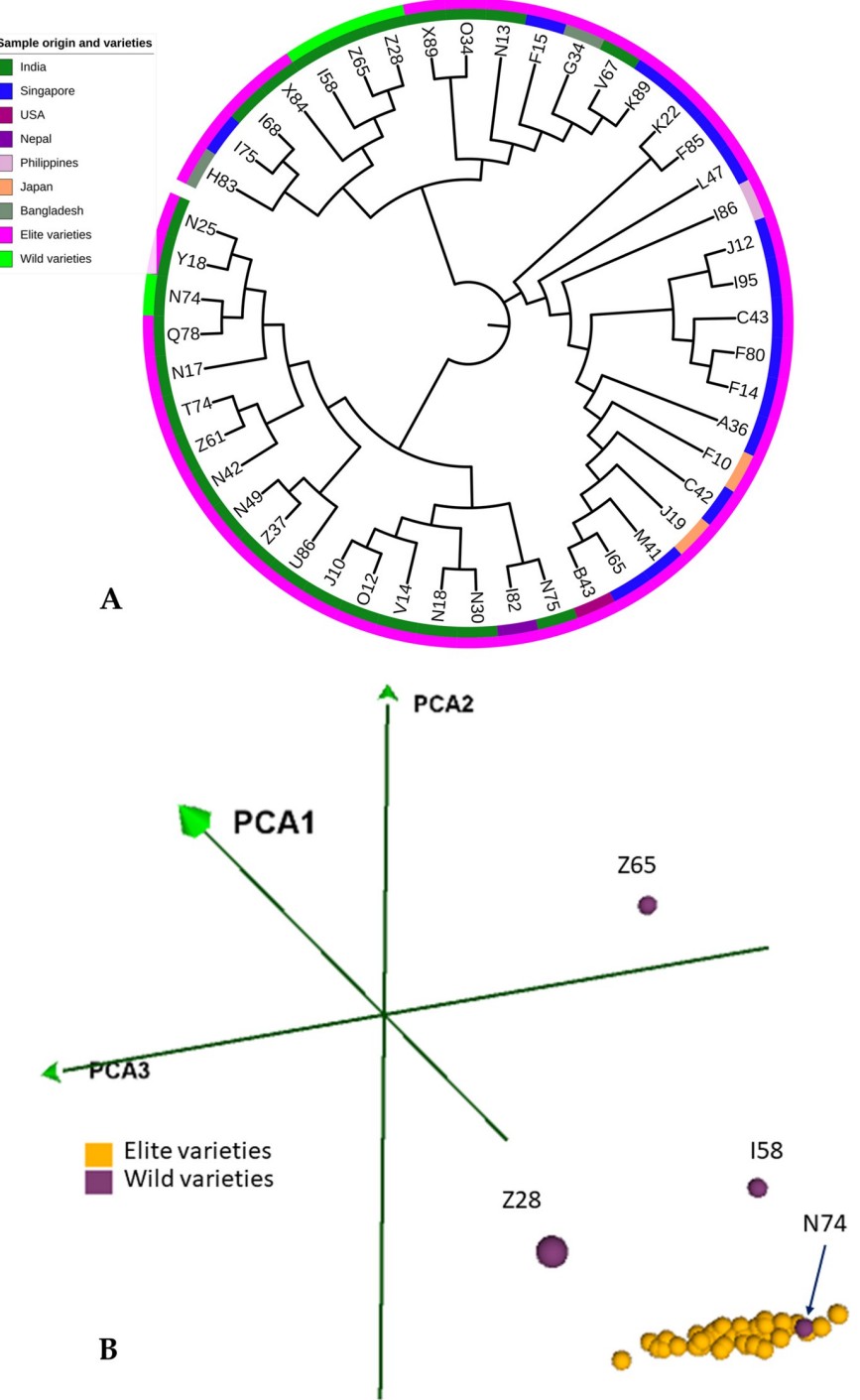

**Fig 6. A.** The combined SNPs set is based on genetic relatedness (NJ tree) between the sesame samples and **B.** PCA analysis.

ddRAD data were generated to call the SNP markers and combined markers used for diversity analysis. In addition to the SNPs, the novel method k-mer sequence based genetic relatedness, distinct k-mer count, k-mer based genetic distance, genic PAV's, heterozygosity, SNPs, Euclidean distance and SNP annotations for representative sample selection.

**Table 1. Sesame distinct samples based on the different criteria.**

| Samples | K-mer | GenePAV | Mash distance | Heterozygosity | Genetic distance | Variant alleles (>3,500) |
|---------|-------|---------|---------------|----------------|------------------|--------------------------|
| X89 | Y | Y | | | | |
| Y67 | Y | Y | | | | |
| Z65 | Y | Y | Y | Y | Y | Y |
| N49 | Y | Y | | | | |
| Y18 | Y | Y | | | | |
| I58 | Y | Y | Y | | Y | Y |
| N74 | Y | Y | Y | | | |
| Z28 | Y | Y | Y | Y | Y | Y |

## K-mer analysis

The reduced representations of the ddRAD-seq data were generated with an average of 149 read depth, which indicated each restriction site was sequenced at multiple folds, causing redundancy. Comparing the genetic sequence between the samples helps in understanding the genetic relationship and the proportion of conserved sequences between the samples. The sequence coverage and the repetitive sequence cause bias in estimating genetic relatedness. To overcome this, the ddRAD-seq sequence reads were split into short k-mers (27 bases length) and called the distinct k-mer for comparison. The distinct k-mers for each sample range from 1.5 million to 5 million k-mers, which indicates the genetic variability with a minimum 1.3 million (B43) to maximum k-mers of 5.5 million (Z37) reported. However, after categorizing the k-mers into the common k-mers and variable k-mers, the Z37, J10, Z28, N74, and F14 samples exhibited the top five highest k-mer variabilities. Additionally, a locality-sensitive hashing technique was used for measuring the k-mer-based genetic distance, which resulted in calculating the pairwise genetic distance between the sesame samples studied. The mash distance reports Z28, I58, J10, N74, Z37 and Z65 are distinct samples that are consistent with earlier k-mer results.

## Data comparison (sdRAD and ddRAD)

We sampled a set of 48 sesame accessions to compare both sdRAD-seq and ddRAD-seq. This analysis provides an opportunity to investigate the source of bias, ease of application and efficiency in terms of SNPs called among both datasets. The approach to analyse the data played an important role in the outcome of the data analysis from each step (from data coverage to SNP count). Assessing both RAD sequence reads coverage on the reference genome showed a significant difference between the sdRAD-seq and ddRAD-seq datasets. A sdRAD-seq sequence data generated with the single digest restriction site enzyme has spanned nearly 3.5% (average of 10.5 Mbp) of the reference genome (Fig 3), whereas the ddRAD-seq has captured only less than one percent (average of 1.4 Mbp). This indicates either the restriction site variability in the genome, i.e; restriction sites used for sdRAD-seq are in high frequency than the restriction sites used for the ddRAD-seq or due to the size selection for library preparation, ie; sdRAD tags have twice more probability than ddRAD tags to have the genome coverage. Such bias in the sd-RAD-seq and ddRAD-seq datasets was also seen in the earlier study [26]. The sequencing read depth for sdRAD-seq on average is 18x, which is much less than the sequence read depth of ddRAD-seq (149 depth) (Fig 3). The read sequencing at higher depth increases the base calling confidence, for example with ddRADseq, on chr1 at 336,353 bp, the I68 sample has 561 reads supporting the A variant genotype (with G as the reference genotype). Similarly, for the same sample, on chr1 chromosome at 318,676 bp has only five reads supporting C

genotypes (with G as the reference genotype), which is of minimum or the required coverage to report a genotype and define it as a variant call. This indicates that the extremely higher sequence depth is not necessary to call the variants. On the other hand, the RADseq technology generates the sequence reads for only the genome-wide restriction sites and such genetic resources enhance understanding of the level of genetic diversity in the sesame population. Even though the higher genome assembly spanning rate was reported for sdRAD-seq data, the ddRAD-seq has predicted more SNPs (27,604) compared to fewer SNPs (13,136) from sdRAD-seq data. This is expected as the ddRAD-seq dataset includes more restriction sites and is expected to be more polymorphic restriction sites than sdRAD-seq [27, 28]. Proportionally, both datasets have a 95% of reference allele as the major allele among the sesame population. Among the sesame samples (sdRAD-seq), Z28 has the most SNP loci with alternative alleles followed by I58 and Z65. Whereas the ddRAD-seq has reported the Z65 sample with most alternative alleles followed by Z28. This indicates that these samples are highly diverse among the sesame population. The SNPs called from both data sets also differ in the density of the SNPs called, as chr4 has the most number of SNPs (1,403) from sdRAD-seq and ddRAD-seq has 2,887 SNPs. SNP comparison between the datasets shows that the sdRAD-seq dataset has a very less number of restriction sites in the intergenic region on the chromosome (low frequency of *ApeKI* restriction sites) than the two restriction sites (*SphI* or *MlucI*) used for ddRAD datasets (Fig 7) (**S2 Fig in S1 File**).

## SNP analysis and heterozygosity

To further understand the nature of genetic variation in the sesame samples, the overall SNPs (combined datasets of sdRAD-seq and ddRAD-seq) demonstrate that 6 samples were having the majority (more than 38,000) of SNPs. The 35,788 average number of locus identified in sesame samples, and diverse samples F14, N30, X89, F15, A36 and N42 have 38,802, 38,383, 38,239, 38,163, 38,079 and 38,034. In addition to the SNPs, the heterozygosity analysis reports 6 samples (Z37, Z28, L47, N75, M41, and Z65) have higher heterozygosity (more than 3,500), of which, except M41, five samples exhibit lower inbreeding coefficient (less than 0), indicating the samples were outcrossed and have more heterozygosity (samples including the wild and elite samples). The variant alleles (more than 3,500) and heterozygosity in I58, Z28 and Z65 sample exhibit higher genetic variations and low inbreeding coefficient in wild samples (Z28 and Z65) compared to the other samples (**S11 Table in S2 File**).

## Evaluation of genetic diversity in sesame samples

We subjected k-mer, heterozygosity and SNP data to genetic diversity analysis and established representative samples from the 48 sesame samples. We detected high levels of genetic diversity in the wild sesame accessions originating from India. For example, Z65 and Z28 samples exhibit a higher level of genetic diversity in the form of the distinct k-mer count, mash k-mer distance, genic variations, and heterozygosity analysis. I58 sample shows the second most diverse sample detected from k-mer count, k-mer distance and euclidean genetic distance. Overall, eight samples (X89, V67, Z65, N49, Y18, I58, N74, Z28) were identified as the most diverse India-origin sample with the distance k-mer count and genic PAV analysis. Of these, four samples (Z65, I58, N74, and Z28) were consistently commonly identified as diverse samples with the mash k-mer diversity analysis. The mash k-mer genetic distance commonly measured for small genomes such as viral [29], microbial [30], whole genome sequence data [31], and also the plant pathogens interactions as studied in *Arabidopsis thaliana* [32]. The level of genetic diversity varies between the different data sets used, the k-mer analysis has identified the 8 samples had the maximum genetic diversity, whereas the polymorphic alleles show that

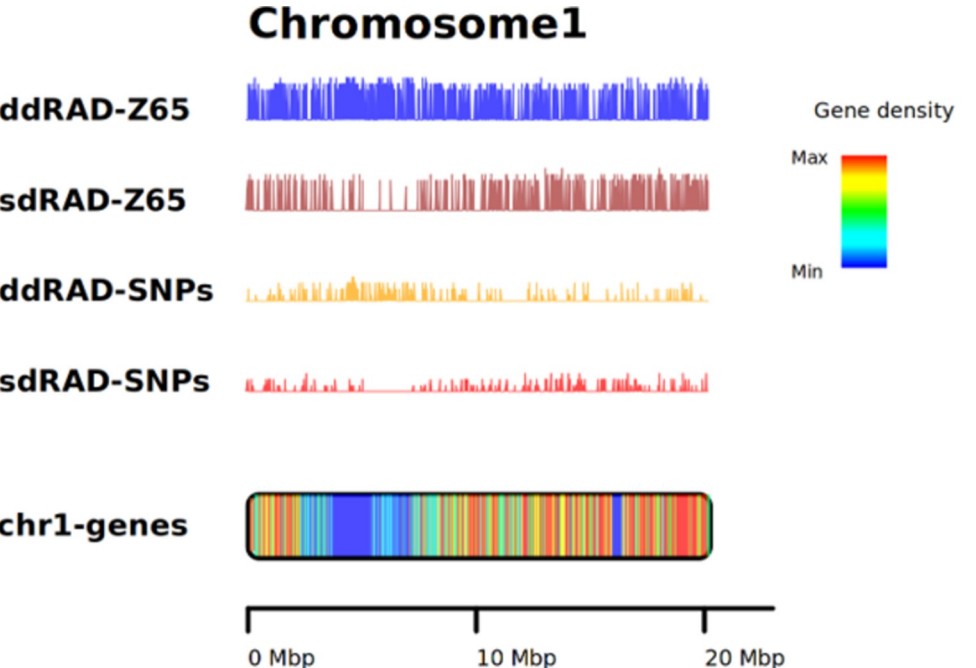

**Fig 7. The sdRAD-seq and ddRAD-seq sequence reads and SNP density comparison.**

the 3 samples (subset of k-mer based diverse samples) have the most number of variants (indicating the most genetic variant samples of the sesame population) (Table 1). The overall genetic diversity from this study varies compared to the whole genome sequence based SNP genetic diversity [21]. Such difference due to the difference in the marker density between ddRAD and whole genome sequence data was earlier reported and comparable [24]. It is advisable to select a representative sample from different origins having higher diversity, however, in this study, only a few samples from non-Indian countries were included which do not exhibit a higher diversity than the sample of Indian origin.

A combination of sdRAD and ddRAD genotype data was used in this study, assessing the data at various levels as k-mer, RAD sequence, gene PAVs and genome-wide genetic distance (mash k-mer distance and NJ). This strategy assessed the genetic diversity of the entire population at different levels. The combination of strategies was earlier used with phenotypic and genotypic analyses to assess the genetic diversity among the wild rice germplasm [33], the oilseed crop, Safflower (*Carthamus tinctorius* L.) core collection was developed with molecular, phenotypic, and geographical diversity [34]. Whereas in olive (*Olea europaea* L.) different molecular markers (DArTs, SSRs, SNPs) and agronomic traits were used [35]. The diverse representative sesame samples was earlier identified with a combination of different parameters, such as a combination of qualitative and quantitative trait descriptors on 2,751 accessions [36], a combination of phenotype and molecular markers on 453 accessions [37], and through combining genetic diversity, traits and agro-ecological type grouping on 4251 accessions [38].

In conclusion, we have generated the sdRAD-seq and ddRAD-seq data and compared the tag sequence mapping rate to assess the data coverage individually. The SNP calls were compared, and the genetic diversity was assessed by combining the variant calling from both datasets. We also identified the diverse sesame samples that hold the genetic variability from SNP level (including the variant allele, heterozygosity, and inbreeding coefficient), to the K-mer sequence and genetic distance analysis. The most diverse sample identified in this study could

be part of the core collection of the sesame germplasm. A similar strategy of defining the core collection can be adapted to a large germplasm collection to assess the diversity in a detailed manner. The combined k-mer and genetic variation used in this study can be adapted to other crop populations. The core collection not only indicates the statistical mean and variances but the range of variability within the population.

## Materials and methods

### Plant material

This study included 48 sesame samples that were genotyped with RAD protocol (sd and dd), of which 26 samples were collected from various locations in India and the remaining 22 samples originated from different countries (S1 Table in S2 File). Before the genotyping experiment, all these sesame accessions were self-crossed for one generation at the Regional Research Station of the Tamil Nadu Agricultural University (TNAU) situated at Virudachalam and the purified seeds were subjected for the genotyping experiment. The seeds were germinated using germination paper towels. Seedlings that were 7–14 days old are used for DNA extraction from fresh tissues (whole seedlings) using DNeasy Plant Kit (Qiagen, USA). The quality and quantity of the extracted DNA were assessed using Qubit fluorometer and electrophoresis.

### RAD-seq data generation

The RAD data generation (both sdRAD-seq and ddRAD-seq) for the DNA of the sesame genotypes was outsourced to AgriGenomics Pvt. Ltd (Hyderabad, India).

The sdRAD-seq data workflow includes the adapters prepared based on the earlier reported protocol [39]. The 1 μg of genomic DNA was digested with *ApeK*I restriction enzyme and P1 P2 adaptors ligated using T4 DNA ligase. Thermo fisher scientific pure link quick gel extraction and PCR purification kit used for pooling and clean-up of the ligated products. The size selection (250–400 bp) was done after 2% agarose gel electrophoresis. PCR amplification was performed to enrich and add the Illumina-specific adapters. QC was checked on the bioanalyzer and final pooling and sequencing were performed on HiSeqX.

The ddRAD data workflow follows a similar protocol as sdRAD-seq workflow applied above but the double digestion of (1 μg) genomic DNA was done with *Sph1* and *MluC1* restriction enzymes [12], and the digested product was cleaned with Ampure beads. The ligation, pooling, size selection, PCR amplification and QC check were done similarly to sdRAD procedure. The final pooling and sequencing were performed on HiSeqX and NovaSeq6000. The pre-processed raw data were subjected to the sd- and dd-RADseq analysis and compared for various parameters.

### RAD-seq analysis

The sdRAD-seq and ddRAD-seq reads were quality trimmed with trimmomatic [40] with low-quality bases (below quality score of 20) and adapters if any were removed, a sliding 4bp window was applied to trim the bases when the average quality score drops below 15, and the remaining clean reads were mapped to the sesame reference genome assembly [5] with Bowtie2 [41]. The basic fastq sequence reads for both the datasets were generated with the in-house developed script (https://github.com/CEG-ICRISAT/Raspberry) and the quality check was performed with fastqc [42] and the results were compiled with multiQC [43]. For each sample, the mapping rate for both RAD-seq was assessed with qualimap [44], Samtools [45] and the variants were called with Stacks pipeline using default parameters [46].

For the k-mer analysis, the cleaned reads subjected to k-mer counting and distinct k-mers were identified with Jellyfish [47]. The k-mer size is 27 nt. The common and unique k-mers were identified based on the presence and absence of a k-mers in 48 sesame samples. The k-mers that appear only once in samples were filtered out as they were likely from the sequencing errors. The k-mer based genetic distance between the 48 samples was measured with Mash [48].

With the above RAD-seq alignments, the gene presence and absence variations between the sesame 48 samples were assessed based on sequence reads coverage mapped to respective genes using a similar method as described earlier [49]. The common (conserved) genes were defined as the genes present in all the accessions, whereas the gene variability identified if a gene missing in one or more accessions. The in-house developed script was used to define the variability from the PAV matrix.

## Genetic diversity analysis

The combined variant calls from both sdRAD-seq and ddRAD-seq datasets were used for the downstream analysis. The SNPs were filtered and plotted to have biallelic SNPs, 0.7 call rate with a minimum maf of 0.1 using the vcftools [50] and CMplot [51]. A 1,000 bootstrap resampling was used to estimate the genetic relationship among the accessions with R "ape" package [52] to construct an NJ tree and visualized in iTOL tree viewer [53].

## Conclusion

In conclusion, we have shown that using different protocols (sdRAD or ddRAD) methods can result in producing different data quantities, coverage and also SNP calls. The variant calls between both protocols were significantly different. The low proportion of common variants between the sdRAD and ddRAD indicates that both protocols are independent and can be used together to have a high density of variants across the genome. Such bias is expected as the source of polymorphic restriction sites, sampling schemes and PCR duplications. The methods to minimize such bias are under development [54] and possibly considered to incorporate into genotyping methods using Bayesian statistics. With the reduced representation, this study shows the possibility to find representative samples with different parameters (SNP, PAV, k-mer, NJ) from the population that act as a source of material to address future challenges in future sesame cultivation.

## Supporting information

**S1 File.**
(DOCX)

**S2 File.**
(XLSX)

**S3 File.**
(DOCX)

## Acknowledgments

The authors are thankful to AgriGenomics Pvt Ltd (Hyderabad, India) for generating the RAD sequencing data using both sdRAD and ddRAD protocols.

## Author Contributions

**Conceptualization:** Rajkumar Subramani, Sunil Archak, Rakesh Singh, Kuldeep Singh, Parimalan Rangan.

**Formal analysis:** Pradeep Ruperao, Prasad Bajaj.

**Funding acquisition:** Rashmi Yadav, Kuldeep Singh, Parimalan Rangan.

**Investigation:** Rajkumar Subramani.

**Methodology:** Prasad Bajaj.

**Project administration:** Rashmi Yadav, Parimalan Rangan.

**Resources:** Vijaya Bhaskar Reddy Lachagari, Sivarama Prasad Lekkala, Ulavappa B. Angadi, Kuldeep Singh.

**Supervision:** Prasad Bajaj, Rajkumar Subramani, Abhishek Rathore, Sean Mayes, Parimalan Rangan.

**Writing – original draft:** Pradeep Ruperao.

**Writing – review & editing:** Pradeep Ruperao, Sean Mayes, Parimalan Rangan.

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
