## [Decision Letter · Decision Letter 0]

23 Mar 2023

PONE-D-23-05086A pilot-scale comparison between single and double-digest RAD markers generated using GBS strategy in sesame (Sesamum indicum L.)PLOS ONE

Dear Dr. Rangan,

Thank you for submitting your manuscript to PLOS ONE. After careful consideration, we feel that it has merit but does not fully meet PLOS ONE’s publication criteria as it currently stands. Therefore, we invite you to submit a revised version of the manuscript that addresses the points raised during the review process.

We look forward to receiving your revised manuscript.

Kind regards,

Tzen-Yuh Chiang

Academic Editor

PLOS ONE

Journal Requirements:

2. Please include a complete copy of PLOS’ questionnaire on inclusivity in global research in your revised manuscript. Our policy for research in this area aims to improve transparency in the reporting of research performed outside of researchers’ own country or community. The policy applies to researchers who have travelled to a different country to conduct research, research with Indigenous populations or their lands, and research on cultural artefacts. The questionnaire can also be requested at the journal’s discretion for any other submissions, even if these conditions are not met.  Please find more information on the policy and a link to download a blank copy of the questionnaire here: https://journals.plos.org/plosone/s/best-practices-in-research-reporting. Please upload a completed version of your questionnaire as Supporting Information when you resubmit your manuscript

Reviewers' comments:

Reviewer's Responses to Questions

**Comments to the Author**

1. Is the manuscript technically sound, and do the data support the conclusions?

Reviewer #1: Partly

Reviewer #2: Partly

2. Has the statistical analysis been performed appropriately and rigorously? 

Reviewer #1: Yes

Reviewer #2: No

3. Have the authors made all data underlying the findings in their manuscript fully available?

Reviewer #1: Yes

Reviewer #2: No

4. Is the manuscript presented in an intelligible fashion and written in standard English?

Reviewer #1: Yes

Reviewer #2: No

5. Review Comments to the Author

Reviewer #1: Line 435. How many basse pair was used for size selection

Line 440. why you were select these enzymes. Do you have any pre-experiment for efficiency of enzymes

Line 442. Size selection. Length of size???

Line 444. What is your length of product. 150 x2 ? and single or paired?

Line 447. What is your parameters in trimmomatic

Line 465. ….genes as described in (Ruperao et al., 2021). Missing sentence???

Line 175…..and ddRAD-seq datasets, respectively

Line 272-273. Please add related references

Line 383-392. There is one study has been conducted sesame for diversity analysis with the use of ddRAD-Seq approach. You should discuss the study

Basak, M., Uzun, B., and Yol, E. (2019). Genetic diversity and population structure of the mediterranean sesame core collection with use of genome-wide SNPs developed by double digest RAD-Seq. PLoS one 14:e0223757. doi: 10.1371/journal.pone.0223757

Line 396. Identifying 8 samples as a core collection is not scientific approach. It is not suitable for core collection idea. Please delete all about core collection

--Discussion is a weak. Please discuss these methods advantages and disadvantages compared to other sequencing approach

Reviewer #2: In their manuscript "A pilote-scale comparison between single and double-digest markers generated using GBS strategy in sesame (Sesamus indicus)", P. Ruperao and colleagues use both single and double digest RAD-sequencing to assess the genetic diversity of multiple sesame strain.

This Work provide interesting genetic resources for sesame cultivators, however some caveats in the approach used and their description maked me think that the manuscript does not reach the standards for scientific publications.

Please find bellow some of the main pitfall I found in the actual version of the manuscript.

Some major comments :

- My first point concern the absence consideration of the fact that rad-sequencing (sd or sd) is highly dependent on the coverage of each samples. This mean that samples with less coverage will de-facto have information, either because some locus were not sequenced, or because the coverage in some locus might not be sufficient to call SNPs.

In the actual manuscript, this aspect is not discuss while it might have a huge impact on the diversity parameter inferred, either with the kmer approach or with the SNPs calling part.

- The introduction claims that genetic information exist for the species, but do not list them (at least if information about populations exist), nor discuss the results of this study in perspectives with previous experiments.

- The choice of the enzymes, that will have a great impact on the number of locus is not explained nor discussed.

- The choice of the strains considered in this study, what is know about their history and how the are related is absent in the actual document and does not allow to grasp the pertinence of the study.

- The choice of the methods chose to “evaluate the genetic diversity” is not explained. Why use a Kmer approach ? How does the number of reads per samples affect the results of this approach ?

- The description of the methods used does not allow the reproducibility of the experiment (Few examples : authors did not describe the size of the fragments selected, nor the parameters for trimming in trimomatic, nor the ones for SNP calling in STACKS)

- Data are not made available, or at least, this is not stated in the manuscript.

-Multiple statement through the document needs references.

Some minor comments :

- Check reference line 431

- Check tense through the document (example line 64: “were” is used wile present tense is used in the rest of the paragraph)

6. PLOS authors have the option to publish the peer review history of their article (what does this mean?). If published, this will include your full peer review and any attached files.

Reviewer #1: No

Reviewer #2: No

---

## [Author Response · Author response to Decision Letter 0]

10 Apr 2023

Dear Editor,

A point by point response to the reviewer comments was submitted in a word document. You may please refer to the response to reviewer comments document for our response and how we have addressed them.

With kind regards

---

## [Decision Letter · Decision Letter 1]

4 May 2023

PONE-D-23-05086R1A pilot-scale comparison between single and double-digest RAD markers generated using GBS strategy in sesame (Sesamum indicum L.)PLOS ONE

Dear Dr. Rangan,

Thank you for submitting your manuscript to PLOS ONE. After careful consideration, we feel that it has merit but does not fully meet PLOS ONE’s publication criteria as it currently stands. Therefore, we invite you to submit a revised version of the manuscript that addresses the points raised during the review process.

We look forward to receiving your revised manuscript.

Kind regards,

Tzen-Yuh Chiang

Academic Editor

PLOS ONE

Reviewers' comments:

Reviewer's Responses to Questions

**Comments to the Author**

1. If the authors have adequately addressed your comments raised in a previous round of review and you feel that this manuscript is now acceptable for publication, you may indicate that here to bypass the “Comments to the Author” section, enter your conflict of interest statement in the “Confidential to Editor” section, and submit your "Accept" recommendation.

Reviewer #1: All comments have been addressed

Reviewer #2: (No Response)

2. Is the manuscript technically sound, and do the data support the conclusions?

Reviewer #1: Yes

Reviewer #2: Partly

3. Has the statistical analysis been performed appropriately and rigorously? 

Reviewer #1: Yes

Reviewer #2: N/A

4. Have the authors made all data underlying the findings in their manuscript fully available?

Reviewer #1: Yes

Reviewer #2: Yes

5. Is the manuscript presented in an intelligible fashion and written in standard English?

Reviewer #1: Yes

Reviewer #2: No

6. Review Comments to the Author

Reviewer #1: (No Response)

Reviewer #2: In their manuscript PONE-D-23-05086R1 P. Ruperao and colleagues proposed a revised version of the manuscript entitled "A pilote-scale comparison between single and double-digest markers generated using GBS strategy in sesame (Sesamus indicus)" comparing performances of single and double digest RAD-sequencing to assess the genetic diversity of multiple sesame strains.

First some side comments:

I would like to point that the version with the tracked change does not track all changes in the manuscript, making the review process harder. Please consider for future submission kipping track of all the modifications.

Also, number of lines pointing to changes does not match the actual changes (i.e “The existing genetic information in the sesame were included in the discussion session of the revised manuscript (Lines 283-300)”, changes are in lines 303 and onward.

Major comments:

In their revised version and answer to comments, the authors answered some of my concerns but some caveats remains.

In my previous review, I was pointing that rad-sequencing (sd or dd) is highly dependent on the coverage of each samples, thus impacting the diversity parameters.

I do understand that the authors used defaults parameters in Stacks, but showing as a supplementary material that the coverage / total number of bp sequenced does not corelates with diversity statics could be useful for the reader to trust your results.

I also suggest to the authors to discuss this issue in the discussion.

I pointed in my first review that the authors does not describe the accessions used un this study. Indeed, authors provide in Sup Tab 1 the location of origin of the accession, but the link between them remains elusive to me. I still think the manuscript would benefit of a brief description of the reasons of inclusion of this list of accessions.

I also think that the manuscript contains languages error (grammar, syntax, vocabulary). I suggest the authors takes the time to thoroughly edit the manuscript.

7. PLOS authors have the option to publish the peer review history of their article (what does this mean?). If published, this will include your full peer review and any attached files.

Reviewer #1: No

Reviewer #2: No

---

## [Author Response · Author response to Decision Letter 1]

9 May 2023

Response to reviewer comments:

Reviewer #1: 

All comments have been addressed.

Reviewer #2: 

Some side comments:

I would like to point that the version with the tracked change does not track all changes in the manuscript, making the review process harder. Please consider for future submission kipping track of all the modifications.

Also, number of lines pointing to changes does not match the actual changes (i.e “The existing genetic information in the sesame were included in the discussion session of the revised manuscript (Lines 283-300)”, changes are in lines 303 and onward..

Response: Apologies for the minor changes in line numbers that had happened due to last minute changes in the manuscript after drafting the response to reviewer document and got slipped to re-edit with correct line numbers. In the present revised manuscript, this is duly taken care of. Thanks for pointing out. 

Major comments:

In their revised version and answer to comments, the authors answered some of my concerns but some caveats remains.

In my previous review, I was pointing that rad-sequencing (sd or dd) is highly dependent on the coverage of each samples, thus impacting the diversity parameters.

I do understand that the authors used defaults parameters in Stacks, but showing as a supplementary material that the coverage / total number of bp sequenced does not corelates with diversity statics could be useful for the reader to trust your results.

I also suggest to the authors to discuss this issue in the discussion.

Response: The coverage of mapped reads was assessed in both vertical coverage (read depth) and horizontal coverage (number of bases captured with sequence reads). The vertical coverage indicates the confidence of the genotype called (represents the quality of called genotype). The horizontal coverage represents the proportion of genome sequence covered (which represents how dense the variants were called, and at the downstream, diversity will be assessed). This is detailed in the discussion in revised manuscript between the line numbers 322-332. 

I pointed in my first review that the authors does not describe the accessions used un this study. Indeed, authors provide in Sup Tab 1 the location of origin of the accession, but the link between them remains elusive to me. I still think the manuscript would benefit of a brief description of the reasons of inclusion of this list of accessions.

Response: In the revised manuscript (lines 102-104), basis of selecting the genotypes was mentioned. Also, in the ‘Supplementary Table 1’ submitted, the required phenotype trait information on basis of which the genotypes were selected was provided. This is now available for readers to have the background information on the choice of the list of accessions used in our study. 

I also think that the manuscript contains languages error (grammar, syntax, vocabulary). I suggest the authors takes the time to thoroughly edit the manuscript.

Response: The revised manuscript was edited for correcting the language error (especially, in the Lines 24, 28, 48, 55, 65, 183, 244, 279-281, 346, 366, 367, 369, 386, 448, 468, 494-496, 503, and 506). 

*******

---

## [Decision Letter · Decision Letter 2]

19 May 2023

A pilot-scale comparison between single and double-digest RAD markers generated using GBS strategy in sesame (Sesamum indicum L.)

PONE-D-23-05086R2

Dear Dr. Rangan,

We’re pleased to inform you that your manuscript has been judged scientifically suitable for publication and will be formally accepted for publication once it meets all outstanding technical requirements.

Kind regards,

Tzen-Yuh Chiang

Academic Editor

PLOS ONE

Additional Editor Comments (optional):

Reviewers' comments:

Reviewer's Responses to Questions

**Comments to the Author**

1. If the authors have adequately addressed your comments raised in a previous round of review and you feel that this manuscript is now acceptable for publication, you may indicate that here to bypass the “Comments to the Author” section, enter your conflict of interest statement in the “Confidential to Editor” section, and submit your "Accept" recommendation.

Reviewer #1: All comments have been addressed

2. Is the manuscript technically sound, and do the data support the conclusions?

Reviewer #1: Yes

3. Has the statistical analysis been performed appropriately and rigorously? 

Reviewer #1: Yes

4. Have the authors made all data underlying the findings in their manuscript fully available?

Reviewer #1: Yes

5. Is the manuscript presented in an intelligible fashion and written in standard English?

Reviewer #1: Yes

6. Review Comments to the Author

Reviewer #1: (No Response)

7. PLOS authors have the option to publish the peer review history of their article (what does this mean?). If published, this will include your full peer review and any attached files.

Reviewer #1: **Yes: **Engin YOL

---

## [Editor Report · Acceptance letter]

23 May 2023

PONE-D-23-05086R2 

A pilot-scale comparison between single and double-digest RAD markers generated using GBS strategy in sesame (*Sesamum indicum* L.) 

Dear Dr. Rangan:

I'm pleased to inform you that your manuscript has been deemed suitable for publication in PLOS ONE. Congratulations! Your manuscript is now with our production department. 

Kind regards, 

on behalf of

Dr. Tzen-Yuh Chiang 

Academic Editor

PLOS ONE